# Pay Better Attention to Attention: Head Selection in Multilingual and Multi-Domain Sequence Modeling

**Hongyu Gong, Yun Tang, Juan Miguel Pino, Xian Li**
Facebook AI Research
{hygong,yuntang,juancarabina,xianl}@fb.com

## Abstract

Multi-head attention has each of the attention heads collect salient information from different parts of an input sequence, making it a powerful mechanism for sequence modeling. Multilingual and multi-domain learning are common scenarios for sequence modeling, where the key challenge is to maximize positive transfer and mitigate negative interference across languages and domains. In this paper, we find that non-selective attention sharing is sub-optimal for achieving good generalization across all languages and domains. We further propose attention sharing strategies to facilitate parameter sharing and specialization in multilingual and multi-domain sequence modeling. Our approach automatically learns shared and specialized attention heads for different languages and domains. Evaluated in various tasks including speech recognition, text-to-text and speech-to-text translation, the proposed attention sharing strategies consistently bring gains to sequence models built upon multi-head attention. For speech-to-text translation, our approach yields an average of $+2.0$ BLEU over 13 language directions in multilingual setting and $+2.0$ BLEU over 3 domains in multi-domain setting.

## 1 Introduction

Recent progress on deep learning models, in particular multi-head attention, has brought significant gains to sequence modeling tasks including speech recognition (Moritz et al., 2020), text-to-text translation (Vaswani et al., 2017), and speech-to-text translation (Vila et al., 2018; Gangi et al., 2019). Attention mechanism allows a model to focus on informative parts of the inputs, and multi-head attention computes attention over inputs by multiple heads independently. With each head attending to different information, multi-head attention potentially captures more complicated data patterns and extracts sophisticated knowledge.

Sequence modeling has attracted a lot of research interest in multilingual and multi-domain settings, where a model is trained on data in multiple language directions and data from different domains respectively. Key advantages of these settings are better data efficiency and the support of knowledge transfer among languages or domains. This is critical for resource-limited scenarios. For example, multilingual translation enhances the performance of low-resource languages via knowledge transfer from high-resource languages (Gu et al., 2018; Inaguma et al., 2019b). Given the data scarcity in individual domains, a common practice is to combine the data from various domains to augment the training set (Wang et al., 2020d). Another appealing aspect of multilingual or multi-domain models is their low deployment and maintenance costs compared with numerous models trained for individual language pairs or domains.

Despite the positive knowledge transfer, negative interference has also been observed in multilingual (or multi-domain) training especially when languages (or domains) are dissimilar. Recent studies reveal from the optimization perspective that conflicting gradients in shared parameters is one cause of interference between languages (or domains) (Yu et al., 2020). A promising direction for interference

mitigation is to design better strategies of parameter sharing. In some previous works, sharing is based on the similarity between languages (or domains), which require expert knowledge or pre-computed relatedness (Wu et al., 2019). Recent studies also propose branches and components specific to languages (or domains) in addition to shared modules (Bapna and Firat, 2019; Guo et al., 2020).

In this work, we bring the mitigation of language and domain interference under a common umbrella, and tackle it by improving parameter sharing within multi-head attention. We propose strategies to select attention heads for different languages or domains. Instead of sharing everything across languages or domains, our model automatically learns to share heads among a subset of languages or domains. It encourages positive transfer within the subset and preserves their specificity without interference from outside the subset. The major contributions of this work are summarized below:

1. We propose attention head selection to mitigate language or domain interference;

2. The parameter sharing strategies are lightweight and preserve inference efficiency;

3. We extensively evaluate attention sharing strategies on various sequence modeling tasks including speech recognition, text-to-text and speech-to-text translation. Consistent gains are achieved across multiple benchmark datasets.

The paper is structured as follows. Section 2 discusses related works on sequence modeling in multilingual and multi-domain setting. In Section 3, we introduce the proposed strategies of head selection in multi-head attention. Section 4 describes the empirical evaluation, followed by a discussion in Section 5. We conclude this paper in Section 6.

## 2 Related Work

**Multilingual learning**. Multilingual modeling has the potential to improve low-resource language performance through knowledge transfer from high-resource languages, and it draws great interest from researchers in speech recognition and translation (Pratap et al., 2020; Heigold et al., 2013; Johnson et al., 2017; Dabre et al., 2020; Liu et al., 2020; Inaguma et al., 2019a; Li et al., 2020). Although impressive progress has been made for low-resource or zero-shot tasks, it is also found the multilingual model has inferior performance on high-resource tasks due to multilingual interference. In order to address this issue, some works focus on multilingual models with task-specific parameters. Different parameter sharing strategies are examined on Transformer (Sachan and Neubig, 2018). Attention dependent on target languages is proposed to enhance multilingual translation (Blackwood et al., 2018). Treating multilingual modeling as an adaptation problem, Bapna and Firat (2019) first build a universal multilingual model for all languages and then finetune newly added adapters for each language pair. Another thread of work is to increase the model capacity to compensate the performance loss in high-resource languages (Pratap et al., 2020). Shazeer et al. (2017) propose mixture-of-experts and select RNN cells based on input tokens. Lepikhin et al. (2020) integrate a mixture of FFN experts in the GShard model, and later Fedus et al. (2021) propose Switch Transformer to route tokens to different FFN sub-layers. Different from previous works, we propose strategies of attention sharing among languages in the level of attention heads for multilingual modeling.

**Multi-domain learning**. Similar to multilingual learning, multi-domain learning (MDL) can effectively utilize data from different domains but also suffers from interference due to inter-domain heterogeneity (Saunders, 2021; Pham et al., 2021). Previous works address this issue from two perspectives: optimization and model architecture. In the optimization aspect, attempts have been made to synchronize the learning speed of different tasks (Chen et al., 2018), adjust the gradients of individual tasks to alleviate gradient conflicts (Yu et al., 2020) and apply regularization to achieve better generalization in different domains (Dakwale and Monz, 2017; Khayrallah et al., 2018; Thompson et al., 2019). In terms of model architecture, domain-specific labels (Kobus et al., 2017), word embedding (Zeng et al., 2018a), sub-networks (Wang et al., 2020d) are adopted to address the issue of domain divergence. The architecture can be specified during the general training with the mixed data from multiple domains (Wang et al., 2020d) or during the finetuning in individual domains (Bapna and Firat, 2019). In this work, we deal with domain interference by leveraging domain-specific attention heads in multi-head attention.

**Attention selection**. Selective self-attention networks propose to apply masking to the inputs and pay more attention to content words (Geng et al., 2020). Liu et al. (2021) select text-related image regions

with attention in multi-modality translation. Compared to these methods, we conduct automatic attention head selection for different tasks and focus on mitigating task interference.

## 3 Model

In this section, we start with preliminaries of multi-head attention, and introduce our approach to attention interference mitigation. We put multilingual and multi-domain sequence modeling under the same umbrella in this study. For the simplicity of the following discussions, we refer to the two settings as multi-task modeling, where a task is one language or one domain. Different from the standard multi-head attention, our model provides more attention heads than those used in computation. Different subsets of heads are assigned to each task so that partial attention sharing enables knowledge transfer and meanwhile mitigates interference. We introduce latent variables to modulate head selection, and propose strategies to learn the head assignment to different tasks.

### 3.1 Preliminary

**Multi-head attention**. As a core component of Transformer, multi-head attention parameterizes each head with key, query and value transformation matrices (Vaswani et al., 2017). The token representation is transformed into key, query and value vectors via these transformations. Each head assigns the attention of this token over the input sequence based on the matching between its query vector and key vectors of other tokens. The value vectors are weighted by the attention as the contextualized token representation. It is passed through linear projection as the output of the attention head. Suppose that head $h$ has output $\mathbf{x}^{(h)}$. Multi-head attention with $H$ heads yields an output $\mathbf{x}$ for the given token, which is the concatenation of all head outputs.

$$\mathbf{x} = \mathbf{x}^{(1)} \oplus \cdots \oplus \mathbf{x}^{(h)} \oplus \cdots \oplus \mathbf{x}^{(H)}, \tag{1}$$

where $\oplus$ is the vector concatenation.

**Interference**. Maximal parameter sharing aims to learn universal knowledge across languages (Wang et al., 2020e) and domains (Zeng et al., 2018b). To capture the task specificity, different languages or domains compete for model capacity, which is observed as the interference in previous studies. The interference results in degraded performance in jointly trained models. However, few works look into the improvement of parameter sharing within multi-head attention. This study explores head selection strategies to mitigate the interference in multilingual and multi-domain models.

### 3.2 Latent Variable for Head Selection

First, we outline our approach to learn a more general-purpose multi-head attention in Transformer from the Bayesian neural network perspective. Suppose that the input sequence is $x$ and the output sequence is $y$. For conditional sequence modeling tasks such as machine translation, the posterior of $p(y \mid x)$ can be computed by marginalizing over the posterior of latent variable $z$, which modulates parameters $\Theta$ in the standard Transformer architecture:

$$p(y \mid x, \Theta) = \mathbf{E}_{p(z|\Theta)}[p(y \mid x, z)] = \int p(y \mid x, z)p(z|\Theta)\,\mathrm{d}z \tag{2}$$

**Parameterization of $z_t$.** In this work, we define $z_t$ as modulating the selection of attention heads by task $t$. We have $z_t = \{z_t^{(h)}\}_h$ where $z_t^{(h)}$ is a discrete latent variable from Bernoulli distribution indicating whether task $t$ selects attention head $h$. This modeling choice allows us to prune attention heads, which preserves computation efficiency as well as regularizes training.

Marginalizing over $z_t$ is intractable given numerous heads in neural models. Therefore, we use variational inference to derive an approximate solution. Suppose that $(x_t, y_t)$ is from task $t$. Specifically, we learn an inference network $q_\phi(z_t)$, which is paramterized with $\phi$, to approximate the true distribution $p(z_t)$ and optimize the evidence lower bound (ELBO) of $p(y|x)$:

$$\log p(y \mid x) \geq \sum_t \left( \mathbf{E}_{q_\phi(z_t)}[\log p_\theta(y_t \mid x_t, z_t)] - \mathrm{KL}(q_\phi(z_t) \parallel p(z_t)) \right), \tag{3}$$

where KL is the KL-divergence between two distributions. In our work, we assume identical probability of each head being selected. Therefore, we have $p(z_t = 1) = \frac{H}{H'}$, where $H$ and $H'$ are numbers of selected attention heads and all head candidates.

**Training and interference.** We use the Gumbel-Softmax reparameterization (Jang et al., 2017) to draw samples of $z_t^{(h)}$ from the posterior $q_\phi(z_t^{(h)})$. It makes the model end-to-end differentiable, while learning discrete policies of head selection without resorting to policy gradients. We adopt a lightweight estimator of $q_\phi(z_t^{(h)})$ by directly learning the logit parameters $\{\phi_t^{(h)}\}$:

$$q_\phi(z_t^{(h)}) = \frac{\exp((\phi_t^{(h)}(1) + \epsilon(1))/\tau)}{\sum_{j \in \{0,1\}} \exp((\phi_t^{(h)}(j) + \epsilon(j))/\tau)} \, , \epsilon \sim \mathcal{G}(0,1) \tag{4}$$

where $\mathcal{G}(0,1)$ is the Gumbel distribution, and $\tau$ is a temperature hyperparameter which increases the discreteness of samples when $\tau \to 0$.

We will discuss different head selection strategies in Section 3.3, which make binary selection decisions based on real-valued posterior $q_\phi(z_t^{(h)})$.

### 3.3 Attention Selection Strategies

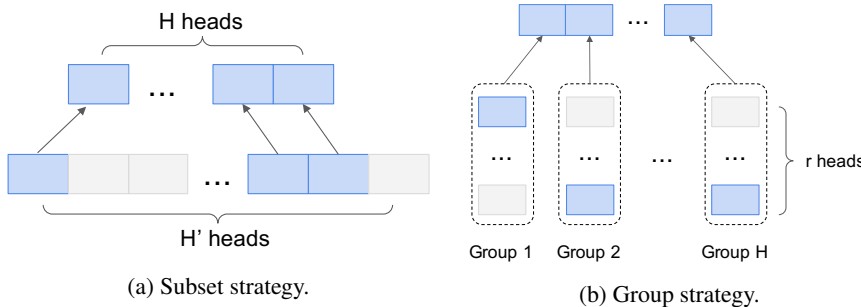

(a) Subset strategy.  (b) Group strategy.

Figure 1: Attention sharing strategies. The blue heads are selected while the grey heads are not.

Suppose that the output dimension of multi-head attention is $d$, and the dimension of each attention head is $\frac{d}{H}$. We provide a large pool of $H'$ ($H' > H$) attention head candidates in every Transformer layer, and $H'$ is a hyperparameter controlling the search space size of attention selection strategies. The model requires attention outputs to have a consistent dimension $d$, so each task needs to select exact $H$ heads among $H'$ candidates. We introduce two strategies for the attention head selection: subset strategy and group strategy.

**Subset strategy**. The subset strategy is straightforward, and we compare the posterior $\{q_\phi(z_t^{(h)}) : h \in [1, H']\}$ of all $H'$ heads given a task $t$. A subset of $H$ heads with the highest posterior are selected by the task, and there are $C_H^{H'}$ subset choices. The subset strategy is described in Fig. 1(a). The binary mask $s_t^{(h)}$ indicates whether an attention head $h$ is assigned to task $t$.

$$s_t^{(h)} = \begin{cases} 1, & h \in \text{TopH}(\{q_\phi(z_t^{(h)})\}), \\ 0, & \text{otherwise}, \end{cases} \tag{5}$$

where $\text{TopH}(\cdot)$ returns the top $H$ heads with the highest values.

The outputs of the selected heads are concatenated as the attention output. Note that the subset strategy does not consider the order of the attention heads. For example, when head 2 and 3 are selected, head 2 contributes to the beginning part of attention output. With head 1 and 2 selected, the output of head 2 goes to the last part of the attention output.

**Group strategy**. We further propose group strategy to preserve the order of attention heads during head selection. Different from the subset strategy, the group strategy first divides $H'$ heads into $H$ groups. As is shown in Fig. 1(b), each group contains $r = \frac{H'}{H}$ candidates. Each task could choose one attention head from each group, and has access to $H$ heads per layer. There are $r^H$ possible combinations of heads. The group strategy keeps the head order in that heads from group $g$ only contribute to $g$'s corresponding dimensions in the attention output. The head with the highest

posterior in its group would be selected by a given task $t$. We use the binary mask $s_t^{(h)}$ to indicate the selection of head $h$ in group $g$.

$$s_t^{(h)} = \begin{cases} 1, & h = \arg\max(\{q_\phi(z_t^{(h')}) : h' \in g\}), \\ 0, & \text{otherwise.} \end{cases} \quad (6)$$

The output of group $g$ is:

$$\mathbf{x}^{(g)} = \sum_{h \in g} s_t^{(h)} \cdot \mathbf{x}^{(h)}. \quad (7)$$

The outputs of $H$ groups are concatenated as the output of the attention module for task $t$.

With either subset or group strategy, the sequence model is trained to assign attention heads to different tasks to maximize the lower bound in inequality (3). The number of additional parameters $\{\phi_t^{(h)}\}$ introduced by our attention selection is only $O(T \times H' \times L)$, where $T$ is the number of tasks, $H'$ is the number of head candidates per layer, and $L$ is the number of layers. It is small compared with the total parameter size of the model, and head selection is thus lightweight and memory efficient. Moreover, the head selection is inherently a pruning process. Regardless of the size of head candidates, only a fixed number of attention heads are involved in computation for a given task. Hence our approach is also computationally efficient in model inference.

## 4  Experiments

We evaluate sequence models in multilingual and multi-domain settings respectively. Various applications are considered including multilingual machine translation (MT), automatic speech recognition (ASR) and speech translation (ST) in both multilingual and multi-domain settings. We integrate attention selection strategies into the self-attention[1] module. Our implementation is based on the FAIRSEQ toolkit (Ott et al., 2019; Wang et al., 2020b). We include widely used sequence models built on multi-head attention as strong baselines below.

1.Transformer (Vaswani et al., 2017). It is a state-of-the-art model in machine translation, which takes texts in source languages as inputs and generates texts in target languages.

2. S2T Transformer (Wang et al., 2020a). As a variant of Transformer for speech processing, S2T Transformer is a stack of a convolutional subsampler and Transformer, where the subsampler processes audio log mel-filter features and sends them to Transformer for text generation.

3. Adapter model (Bapna and Firat, 2019). Adapters have been shown as an effective approach to language and domain adaptation. Task-specific layers are added on top of each Transformer layer in a well-trained (S2T) Transformer. A typical adapter layer consists of two feed-forward sub-layers.

4. Static strategy of head selection. A static strategy assigns each task with a fixed subset of attention heads based on the task similarity (Standley et al., 2020; Sen et al., 2019). In the multilingual setting, we group languages into linguistic families, and each family is assigned with an exclusive set of heads. As for the multi-domain setting, each domain has its own set of attention heads.

We report parameter size and decoding speed as memory and computation efficiency metrics respectively. Decoding speed is measured by the number of tokens decoded per second by one GPU.

### 4.1  Machine Translation

The task of machine translation is to translate a text from one language to another. The metric BLEU measures the overlap between model translations and the ground truth (Papineni et al., 2002).

**Dataset**. We experiment with public multilingual machine translation datasets collected by WMT shared tasks as used by (Liu et al., 2020). The dataset consists of parallel sentences between English and other 14 languages[2]. Its data statistics are summarized in Appendix A.1. We evaluate models on

---

[1]We also tried head selection in the encoder-decoder attention but did not observe big improvements when using it alone or in combination with self-attention head selection.

[2]The 14 languages are: Chinese (zh), Czech (cs), Estonian (et), Finnish (fi), French (fr), German (de), Gujarati (gu), Kazakh (kk), Latvian (lv), Lithuanian (lt), Romanian (ro), Russian (ru), Spanish (es), Turkish (tr).

both one-to-many (O2M) and many-to-one (M2O) translations, which are translation from English to 14 languages and from 14 languages to English respectively.

**Model configurations**. The attention selection is based on the source language on the encoder side for M2O translation, and is based on the target language in the decoder part for O2M translation. For both subset and group strategies, the number of attention head candidates is set as 8 in each layer (i.e., H'=8), and only 4 heads (i.e., H=4) are selected for computation. We will discuss how the hyperparameter $H'$ affects model performance in Section 5. For static strategy, we group languages into 5 linguistic families[3]. Each family is assigned with 4 attention heads which are shared by all languages in this family. Therefore, a total of 20 attention heads are used in the static strategy.

Other baselines have 4 attention heads in each Transformer layer. We also include a Transformer baseline with 8 attention heads, which measures the effect of increased attention heads. All models have 6 encoder layers and 6 decoder layers, the embedding dimension is 512 and the feed-forward dimension is 1024. They are trained with a batch size of 131k tokens and a learning rate of 0.0007. For O2M translation, attention selection models and Transformer are trained for 140k steps. As for M2O translation, they are trained for 100k steps. The adapter model is initialized with parameters from the trained Transformer, and tunes adapter layer parameters for 40k steps with Transformer parameters frozen. Adapter layers are added to Transformer for each language direction, and they have an intermediate dimension of 256. The dimension is selected so that the number of parameters (460M) in the adapter model is close to the parameter size (420M) in attention selection models.

Table 1: BLEU (↑) of Machine Translation on WMT Datasets (#Params: the number (Million) of model parameters. Speed: the number of tokens decoded per second. AVG-A: average BLEU over 14 directions, High and Low are average BLEU over high- and low-resource languages respectively.)

| | O2M | | | | | M2O | | | | |
| | #Params | Speed | BLEU | | | #Params | Speed | BLEU | | |
| | (M) | (tok/s) | AVG-A | High | Low | (M) | (tok/s) | AVG-A | High | Low |
|---|---|---|---|---|---|---|---|---|---|---|
| Transformer (H=4) | 416 | 1140 | 20.1 | 25.7 | 16.0 | 416 | 1252 | 22.8 | 27.9 | 19.0 |
| Transformer (H=8) | 416 | 1089 | 20.6 | 26.7 | 16.1 | 416 | 1156 | **23.7** | **29.0** | **19.7** |
| Adapter | 460 | 1021 | 20.9 | 26.7 | **16.6** | 460 | 1117 | 23.3 | 28.7 | 19.3 |
| Static strategy | 434 | 1133 | 20.9 | **27.1** | 16.3 | 434 | 1250 | 23.6 | **29.0** | 19.5 |
| Group strategy | 420 | 1137 | **21.0** | 27.1 | 16.4 | 420 | 1245 | 23.5 | 28.8 | 19.6 |
| Subset strategy | 420 | 1133 | 20.9 | 27.0 | 16.4 | 420 | 1250 | 23.3 | 28.7 | 19.4 |

**Results**. We group 14 language directions based on their amount of training data. We have 6 high-resource languages with more than 10M parallel sentences, and 8 low-resource languages with fewer than 10M sentence pairs. Table 1 shows model performance on WMT datasets. More attention heads improve Transformer performance while hurting the decoding speed. In comparison with Transformer with 4 heads, group strategy achieves +0.9 and +0.7 BLEU on average of 14 language directions in O2M and M2O translations respectively at a comparable decoding speed. Transformer with 8 heads and adapter achieve BLEU scores comparable to both group and subset strategies but fall behind in inference efficiency. Static strategy demonstrates comparable performance to group and subset strategies in all metrics except the parameter size.

## 4.2 Speech Recognition

The task of Automatic Speech Recognition (ASR) is to transcribe source audios in the same language. Word error rate (WER) is ASR evaluation metric, which measures the difference of model outputs from the ground truth (Klakow and Peters, 2002). Lower WER indicates better recognition.

**Model configuration**. Models included in the experiments of speech recognition are S2T Transformer, S2T Transformer with adapter layers, S2T Transformer with static, group and subset strategies. With static strategy, we group 8 languages into 2 families[4], and each family has an exclusive set of 4 attention heads. Following the setup of (Salesky et al., 2021), all models have 1024 channels in the input convolutional subsampler, 12 encoder layers and 6 decoder layers with 4 attention heads per layer. Again we include the S2T Transformer baseline with 8 heads. The embedding dimension is 256 and the feed-forward dimension is 2048. We set a batch size of 320k tokens and a learning

---

[3](1) Indo-European family: cs, de, es, fr, gu, lt, lv, ro and ru; (2) Estonian family: et; (3) Uralic family: fi; (4) Turkic family: kk and tr; (5) Sino-Tibetan family: zh.

[4](1) Afro-Asiatic family: ar; (2) Indo-European family: de, el, es, fr, it, pt and ru.

rate of 0.0005 during training. Attention selection models and S2T Transformer are trained for 250 epochs. Adapter model is initialized with parameters of the trained S2T Transformer, and is then trained for another 200 epochs with only adapter layer parameters tuned. The intermediate dimension of adapter layers is again set as 256. To prevent over-fitting, we stop the model training when the model does not improve on the validation set for 10 epochs. To reduce the performance variance, we average checkpoints of the last 10 epochs, and use the averaged model for evaluation.

### 4.2.1 Multilingual Speech Recognition

**Dataset**. We use the multilingual TEDx (mTEDx) dataset for speech recognition (Salesky et al., 2021). It collects audio recordings from TEDx talks. Eight languages are covered including Arabic (ar), German (de), Greek (el), Spanish (es), French (fr), Italian (it), Portuguese (pt) and Russian (ru).

Table 2: WER (↓) of Speech Recognition on mTEDx Dataset

|  | #Params (M) | Speed (tok/s) | BLEU | | | | | | | | |
|---|---|---|---|---|---|---|---|---|---|---|---|
|  |  |  | AVG | ar | de | el | es | fr | it | pt | ru |
| S2T Transformer (H=4) | 31M | 1118 | 49.0 | 109.5 | 72.3 | 43.3 | 23.9 | 27.8 | 28.6 | 31.0 | 55.3 |
| S2T Transformer (H=8) | 31M | 1052 | 46.0 | 103.3 | 69.7 | 40.5 | 21.5 | 25.4 | 25.7 | 28.0 | 53.5 |
| Adapter | 50M | 1016 | 41.1 | 93.4 | 57.2 | 33.0 | 21.4 | 25.3 | 24.3 | 27.2 | 46.7 |
| Static strategy | 35M | 1108 | 49.4 | 110.1 | 72.9 | 43.8 | 24.1 | 27.9 | 29.4 | 31.4 | 55.8 |
| Group strategy | 35M | 1107 | **40.0** | 94.2 | 59.8 | 33.5 | 18.2 | 22.0 | 21.9 | 24.6 | 45.5 |
| Subset strategy | 35M | 1114 | 44.7 | 97.3 | 65.3 | 38.7 | 22.4 | 25.8 | 26.4 | 29.0 | 52.4 |

**Results**. S2T transformer share all parameters among languages. Attention selection models select attention heads based on the source and target languages. Adapter adds adapter layers based on the language directions. We report the ASR results in Table 2. It brings down $6.1\%$ WER for S2T Transformer to increase from $4$ to $8$ attention heads. Compared to S2T Transformer with $4$ heads, adapter model reduces the WER by $16.1\%$ and subset strategy by $8.8\%$, while static strategy does not change the performance too much. Group strategy achieves the largest drop of $18.4\%$ in WER of S2T Transformer (H=4) with comparable decoding speed. Moreover, it outperforms S2T Transformer with 8 heads in both speed and WER.

### 4.2.2 Multi-Domain Speech Recognition

**Dataset**. Besides mTEDx data, we include two other public datasets, CoVoST 2 and EuroParl, which are commonly used for speech translation. Since source audios are accompanied by transcripts, we could use their source audio-text data for speech recognition tasks. We investigate multi-domain modeling with these three datasets.

1. CoVoST 2 (Wang et al., 2020c). With Common Voice as the audio source, CoVoST 2 covers speech-to-text translations from 21 languages to English and from English to 15 languages.
2. EuroParl (Iranzo-Sánchez et al., 2020). It provides paired audio-text instances from and into 6 European languages, which are compiled from the debates in European Parliament.

Table 3: WER (↓) of Speech Recognition on mTEDx, CoVoST 2 and EuroParl Dataset

|  | #Params (M) | Speed (tok/s) | BLEU | | |
|---|---|---|---|---|---|
|  |  |  | mTEDx | CoVoST 2 | EuroParl |
| Separate S2T Transformers (H=4) | 32M | 1378 | 49.0 | 41.9 | 115.0 |
| Separate S2T Transformers (H=8) | 32M | 1280 | 46.0 | 40.7 | 94.0 |
| Joint S2T Transformer (H=4) | 32M | 1409 | 42.7 | 38.3 | 25.6 |
| Joint S2T Transformer (H=8) | 32M | 1299 | 43.3 | 38.6 | 25.9 |
| Adapter | 39M | 1274 | 41.7 | 37.0 | **24.0** |
| Static strategy | 39M | 1402 | 46.3 | 41.6 | 30.0 |
| Group strategy | 36M | 1400 | **41.0** | **36.4** | 24.3 |
| Subset strategy | 36M | 1403 | 41.8 | 37.0 | 25.0 |

**Results**. In the multi-domain setting, attention selection models assign different heads to each domain. The static selection strategy provides each domain with an exclusive subset of attention heads. Adapter model adds domain-specific adapter layers to S2T Transformer. Table 3 reports WER of models trained for 400 epochs in three domains: mTEDx, CoVoST 2 and EuroParl respectively. The S2T Transformer jointly trained on multi-domain data (in the row of "Joint S2T Transformer

(H=4)") reduces WER by 12.9%, 8.6% and 77.7% in three domains respectively, when compared with the models separately trained in individual domains (in the row of "Separate S2T Transformer (H=4)"). This demonstrates the benefits of positive transfer between domains.

The performance of speech recognition could be further improved by the mitigation of the domain interference. Attention selection with group and subset strategies outperform that with the static strategy. Both attention selection and adapter model achieve lower WER than the joint S2T Transformer with both 4 and 8 attention heads. Attention selection with group strategy has the lowest WER on both mTEDx and CoVoST 2 datasets, decreasing WER by 4.0% and 5.0% respectively in comparison with joint S2T Transformer (H=4). The best system on EuroParl is adapter model, yielding a WER reduction by 6.3% than the joint S2T Transformer (H=4).

## 4.3 Speech Translation

Now with a focus on the task of speech translation, we again design experiments in multilingual and multi-domain settings. In the multilingual setup, we train translation models with samples in multiple languages to investigate language interference. As for the multi-domain setup, the models are trained with data from multiple domains so that we could look into the domain interference. BLEU serves as the evaluation metric of speech translation systems.

**Baselines**. We use the same baselines as in speech recognition. As recommended by (Salesky et al., 2021), we initialize the encoders in speech translation with the encoders trained in the task of speech recognition in Section 4.2 for the purpose of improving training efficiency and performance.

**Model configurations**. All models are trained for up to 400 epochs. Other model configurations in ST are the same as those in ASR.

### 4.3.1 Multilingual Speech Translation

To explore language interference, we perform experiments on multilingual speech translation.

**Dataset**. We again use mTEDx dataset for multilingual speech translation. Besides speech recognition data, mTEDx also collects speech translation data from TEDx talks. Its test set covers 13 language directions. The training data is provided in 10 of these directions, so there are 3 zero-shot directions.

Table 4: BLEU (↑) of Speech Translation on mTEDx (AVG-A: average over all directions, AVG-T: average of 10 training directions, and AVG-Z: average of 3 zero-shot directions)

| | #Params (M) | Speed (tok/s) | BLEU AVG-A | AVG-T | AVG-Z |
|---|---|---|---|---|---|
| S2T Transformer (H=4) | 31M | 1038 | 13.2 | 14.6 | 8.5 |
| S2T Transformer (H=8) | 31M | 968 | 13.7 | 15.1 | 9.0 |
| Adapter | 55M | 826 | - | 14.8 | - |
| Static strategy | 33M | 1030 | 13.2 | 14.5 | 8.8 |
| Group strategy | 35M | 1024 | **15.2** | **16.7** | **10.4** |
| Subset strategy | 35M | 1033 | 13.3 | 14.7 | 8.5 |

**Results**. Here the static strategy of head selection groups source languages into two families as in multilingual ASR task, and all target languages fall into the same Indo-European family. Table 4 summarizes the multilingual speech translation results on mTEDx. S2T Transformer has +0.4 BLEU with heads increased to 8. Since adapter model brings in language-specific layers, it cannot deal with zero-shot translations. Group strategy, subset strategy and adapter model bring improvements over S2T Transformer (H=4) which are jointly trained in 13 language directions. It suggests that multiple languages interfere within S2T Transformer whose parameters are shared by all languages. Attention selection with group strategy achieves the best translation performance. In comparison with S2T Transformer (H=4), group strategy achieves an average of +2.1 and +1.9 BLEU in training and zero-shot directions respectively. It leads to +2.0 BLEU on average of all directions.

### 4.3.2 Multi-Domain Speech Translation

In this experiment, we investigate interference across domains in the task of speech translation, and evaluate the effectiveness of different models in multi-domain training. The attention selection now is

based on the data domain instead of languages, i.e., samples in different domains would choose their own attention heads. Similarly for adapter model, its adapter layers are domain-specific in this setup.

We again use CoVoST 2 and EuroParl as additional domains. We focus on the 13 language directions in mTEDx test set, and use the subset of CoVoST 2 and EuroParl corpora in the same directions. CoVoST 2 has 5 common directions[5] and EuroParl has 11 common directions[6] as mTEDx. Details about these datasets are included in Appendix A.1.

Table 5: BLEU ($\uparrow$) of Speech Translation on mTEDx, CoVoST 2 and EuroParl Dataset

| | #Params (M) | Speed (tok/s) | TEDx | | | CoVoST 2 | EuroParl |
| | | | AVG-A | AVG-T | AVG-Z | AVG | AVG |
|---|---|---|---|---|---|---|---|
| Separate S2T Transformer (H=4) | 32M | 1105 | 13.2 | 14.6 | 8.5 | 17.6 | 19.1 |
| Separate S2T Transformer (H=8) | 32M | 957 | 13.7 | 15.1 | 9.0 | 19.0 | 19.0 |
| Joint S2T Transformer (H=4) | 32M | 1148 | 13.7 | 13.6 | 13.9 | 17.3 | 19.0 |
| Joint S2T Transformer (H=8) | 32M | 1062 | 13.7 | 13.6 | 13.9 | 17.4 | 19.1 |
| Adapter | 39M | 940 | 14.0 | 14.3 | 13.2 | 17.9 | 20.0 |
| Static strategy | 39M | 1138 | 11.5 | 12.0 | 9.7 | 14.5 | 16.9 |
| Group strategy | 36M | 1141 | **15.6** | **15.9** | **14.8** | **19.6** | **20.8** |
| Subset strategy | 36M | 1145 | 13.8 | 14.3 | 13.1 | 17.9 | 19.2 |

**Results**. Table 5 shows the average BLEU of speech translation in mTEDx, CoVoST 2 and EuroParl. We report results in rows of "Joint S2T Transformer" when S2T Transformers are trained with the mixture of three datasets. The results are included in rows "Separate S2T Transformer" when S2T Transformers are trained on each dataset independently. Zero-shot translations in mTEDx benefit a lot from additional data of CoVoST 2 and EuroParl, as the joint S2T Transformer (H=4) shows an average of +5.4 BLEU over separate S2T Transformer (H=4). However, there is a drop of 1.0 BLEU in its training directions, brought by the interference from CoVoST 2 and EuroParl domains.

Again we observe that static strategy falls behind group and subset strategies. Attention selection with learned strategies and adapter model bring gains to the joint model in individual domains. Compared with the joint S2T Transformer (H=4), adapter model improves mTEDx translation by 0.3 BLEU, CoVoST 2 translation by 0.6 BLEU and EuroParl by 1.0 BLEU on average. The attention selection with group strategy outperforms all other models. Its average BLEU gain over adapter model is 1.6 BLEU in mTEDx, 1.7 BLEU in CoVoST 2 and 0.8 in EuroParl.

## 5 Discussion

**Hyperparameter $H'$**. The attention selection models set a hyperparameter $H'$ as the total number of attention head candidates in multi-head attention, which controls the search space of attention sharing strategies. We now explore how the performance varies with $H'$ for group and subset strategies.

Evaluated on the task of multilingual speech recognition, models have the same hyperparameters as those in multilingual ASR experiments except for $H'$. Attention selection models are configured with $H' = 4, 8, 12, 16$ respectively, and Figure 2 shows the change of WER with $H'$.

When $H' = 4$, there is no attention selection and all attention heads are shared by different languages. We observe a large drop of error rate as $H'$ increases from 4 to 8. For the subset strategy, WER keeps decreasing when the number of head candidates grows from 4 to 16. As for group strategy, $H' = 8$ is the optimal hyperparameter on the ASR task. As we continue

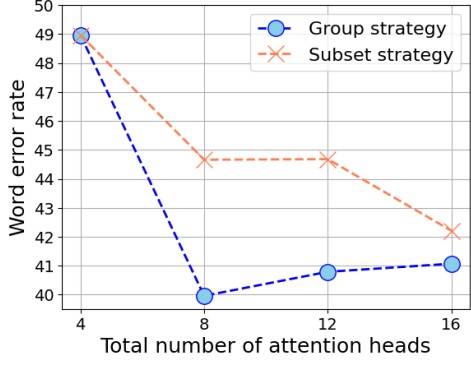

Figure 2: WER of speech recognition on mTEDx with different $H'$.

[5] {es, fr, it, pt, ru}-en
[6] es-{en, fr, it, pt}, fr-{en, es, pt}, it-{en, es}, pt-{en, es}

increasing $H'$ to 12 and 16, the error rate increases a bit. The performances of subset and group strategies are close when $H' = 16$.

The search space of group strategy is a strict subset of the space of subset strategy. However, we observe that group strategy shows comparable or better performance than subset strategy across tasks, including MT, ASR and ST. One possible explanation is that group strategy keeps the head order information while subset strategy does not. With a larger pool of head candidates, there is less sharing among tasks. The performance of the group strategy degrades a bit due to less positive transfer dependent on attention sharing. As for the subset strategy, better head assignments are learned in the enlarged search space.

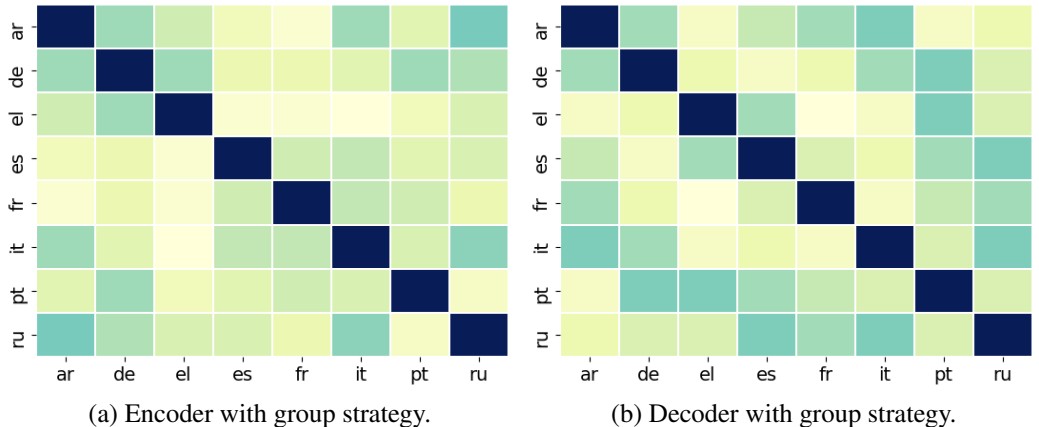

(a) Encoder with group strategy.          (b) Decoder with group strategy.

Figure 3: Heatmap to visualize the sharing between languages in multilingual ASR (The darker a language pair is, the more attention heads they share.)

**Attention Sharing among Languages**. We now analyze the attention sharing pattern among languages. Take the multilingual model on mTEDx speech recognition as an example, whose head selection is learned with group strategy. We count the number of heads shared by each language pair in the model, and visualize it with a heatmap in Fig. 3, where the darkness reflects the amount of sharing. The diagonal cells in the heatmap corresponds to the number of attention heads used by each language, i.e., the total number of attention heads in all layers.

For European languages including Spanish (es), French (fr), Italian (it) and Portuguese (pt), their shared attention heads are fewer in decoder than in encoder. This seems contradicted with previous findings that parameter sharing is beneficial for languages with high linguistic proximity. We note that they are high-resource languages in mTEDx corpus, which is also justified by their relatively lower WER. Their data is sufficient to learn good speech recognition, and sharing parameters with other languages hurt the preservation of the language specificity. This explains why the high-resource European languages do not share too many heads in the learned group strategy.

Another pattern we observe from Fig. 3 is that low-resource languages tend to share more attention heads with high-resource languages. For example, Arabic (ar) and Russian (ru) have relatively more sharing with Italian (it) than other languages. Low-resource languages benefit from the knowledge transfer from high-resource ones. Due to page limit, we include more discussions in Appendix A.3.

## 6   Conclusion

Research efforts in multilingual and multi-domain modeling have been driven by the increasing need to improve data efficiency and model performance. In this work, we propose head selection strategies to allow attention heads to be shared or specialized for different languages or domains. It effectively mitigates interference within multi-head attention which is a core part of strong sequence models, and demonstrates good empirical gains in various text generation tasks.

This work has several limitations left for future research. We did not explore head selection based on both language and data domain. We did not analyze model fairness and robustness. As a technology used for text generation, the model might have systemic bias or produce inappropriate outputs.

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
