# A  Appendix

## A.1  Dataset

Table 6 summarizes the number of parallel sentences for 14 languages in WMT shared tasks. Table 7 covers three datasets, CoVoST 2, EuroParl and mTEDx on ASR task, and reports their number of utterances in 8 languages. Table 8 shows data sizes of three ST datasets including CoVoST 2, EuroParl and mTEDx. It reports the number of utterances in 13 language directions.

Table 6: Data Statistics of WMT Datasets

| Language | Code | Size | Language | Code | Size |
|---|---|---|---|---|---|
| Gujarati | gu | 10k | Kazakh | kk | 91k |
| Turkish | tr | 207k | Romanian | ro | 608k |
| Estonian | et | 1.94M | Lithuanian | lt | 2.11M |
| Finnish | fi | 2.66M | Latvian | lv | 4.50M |
| Czech | cs | 11M | Spanish | es | 15M |
| Chinese | zh | 25M | German | de | 28M |
| Russian | ru | 29M | French | fr | 41M |

Table 7: Data Statistics of Speech Recognition Task (# of Utterances)

| Data | CoVoST 2 | | | EuroParl | | | mTEDx | | |
|---|---|---|---|---|---|---|---|---|---|
| Split | Train | Dev | Test | Train | Dev | Test | Train | Dev | Test |
| ar | 2,283 | 1,758 | 1,695 | - | - | - | 11,442 | 1,079 | 1,066 |
| de | 127,577 | 13,503 | 13,503 | 13,099 | 2,653 | 2,644 | 6,659 | 1,172 | 1,126 |
| el | - | - | - | - | - | - | 12,521 | 982 | 1,027 |
| es | 78,958 | 13,203 | 13,204 | 7,537 | 1,951 | 1,831 | 99,660 | 905 | 1,012 |
| fr | 207,286 | 14,755 | 14,750 | 13,006 | 1,593 | 1,848 | 114,488 | 1,036 | 1,059 |
| it | 31,638 | 8,877 | 8,892 | 11,649 | 1,414 | 1,763 | 48,089 | 931 | 999 |
| pt | 9,158 | 3,315 | 4,021 | 4,977 | 1,794 | 2,292 | 88,123 | 1,013 | 1,020 |
| ru | 12,112 | 6,110 | 6,300 | - | - | - | 28,627 | 973 | 1,132 |

Table 8: Data Statistics of Speech Translation Task (# of Utterances)

| Data | CoVoST 2 | | | EuroParl | | | mTEDx | | |
|---|---|---|---|---|---|---|---|---|---|
| Split | Train | Dev | Test | Train | Dev | Test | Train | Dev | Test |
| el-en | - | - | - | - | - | - | 4,215 | 938 | 1,024 |
| es-en | 78,958 | 13,203 | 13,204 | 7,403 | 1,947 | 1,816 | 35,186 | 899 | 1,001 |
| es-fr | - | - | - | 4,673 | 1,115 | 1,082 | 3,549 | 904 | 1,005 |
| es-it | - | - | - | 4,476 | 1,065 | 1,079 | 5,530 | 16 | 262 |
| es-pt | - | - | - | 4,727 | 1,141 | 1,089 | 20,467 | 898 | 1,002 |
| fr-en | 207,286 | 15,560 | 14,952 | 12,446 | 1,481 | 1,804 | 29,634 | 1,035 | 1,058 |
| fr-es | - | - | - | 7,857 | 1,072 | 1,098 | 20,407 | 1,034 | 1,057 |
| fr-pt | - | - | - | 8,183 | 1,048 | 1,100 | 13,047 | 1,035 | 1,058 |
| it-en | 31,638 | 9,095 | 8,937 | 11,285 | 1,400 | 1,686 | - | 929 | 999 |
| it-es | - | - | - | 6,614 | 877 | 885 | - | 929 | 999 |
| pt-en | 9,158 | 3,590 | 4,254 | 4,918 | 1,747 | 2,286 | 29,940 | 1,002 | 1,019 |
| pt-es | - | - | - | 3,132 | 1,218 | 1,256 | - | 1,001 | 1,018 |
| ru-en | 12,112 | 9,497 | 8,634 | - | - | - | 4,829 | 970 | 1,124 |

**Data license**. The machine translation data released for WMT shared tasks can be freely used for research purposes. The multilingual TEDx corpus is released under a CC BY-NC-ND 4.0 license, and can be freely downloaded. CoVoST 2 data is released under CC0 license. As for EuroParl, it is released under a Creative Commons license, and it is freely accessible and downloadable.

## A.2  Experiment

The experiments were performed in the internal cluster. For machine translation experiments, we used 32 GPUs and each model was trained for around 3 days. On the task of speech recognition,

models are trained on 8 GPUs. It took approximately 1 day for models to converge in multilingual setting, and 2 days in multi-domain setting. On the task of speech translation, the training time was also 1 day for multilingual models, and 2 days for multi-domain models. Speech translation models were trained with 8 GPUs.

## A.3 Discussion

In this section, we provide insights into learned attention head selection with further result analysis.

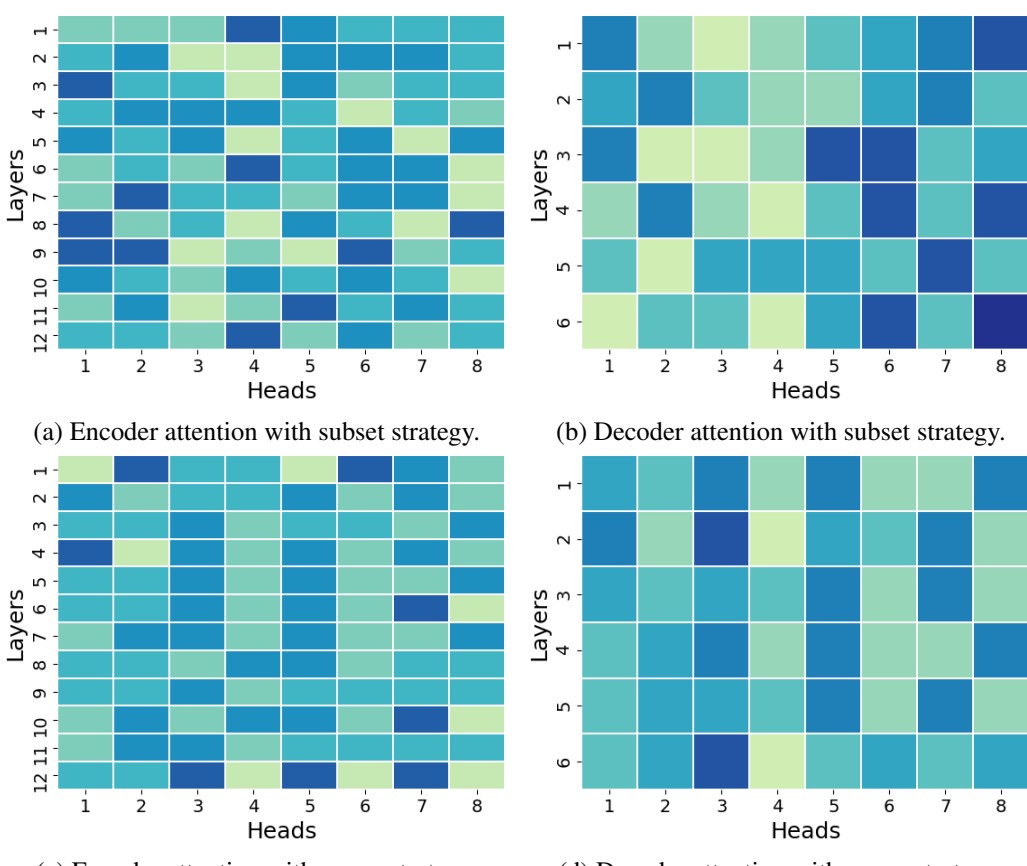

(a) Encoder attention with subset strategy.

(b) Decoder attention with subset strategy.

(c) Encoder attention with group strategy.

(d) Decoder attention with group strategy.

Figure 4: Heatmap to visualize the load of attention heads (The darker a head is, the more languages it supports).

**Load of Attention Heads**. To gain an insight into the load of attention heads, we analyze how many languages an attention head is used by. With both subset and group strategies, we look into attention heads in encoder and decoder respectively. We study ASR models which learn language based attention selection on mTEDx data covering 8 languages. The language load of each attention head is measured by the number of languages sharing the given head. Fig. 4 visualizes the load of attention heads in each layer with a heatmap. The darkness reflects the load of an attention head.

By comparing encoder heads in Fig. 4(a) and (c), we note that group strategy results in more balanced load among attention heads than subset strategy, as there is less color variation in the heatmap of group strategy. Similar pattern could be observed in decoder, and decoder attention heads have more balanced load with group strategy.

Now we compare the load of attention heads across layers. With subset strategy, the load imbalance is observed in heads of almost every encoder and decoder layer from the color contrast in the heatmap. As for group strategy, the load is more balanced in heads of middle layers (i.e., encoder layers $5 - 9$ and decoder layers $3 - 5$) than those in bottom and top layers.

**Head Selection in Encoder and Decoder**. In our experiments, attention selection is applied to both encoder and decoder in ASR and ST experiments, considering that both encoder and decoder handle multiple languages. We want to measure how the model performance is affected by attention selection in encoder and decoder respectively. Taking the multilingual ASR as an example, Table 9 reports WER of models which enable attention selection in encoder only, in decoder only as well as in both encoder and decoder. We set the same hyperparameters as used in the experiment of multilingual ASR. When the attention selection is applied to encoder (or decoder) only, 4 attention heads are shared by all languages in each decoder (or encoder) layer.

Table 9: Ablation Study in WER ($\downarrow$) of Multilingual Speech Recognition on mTEDx

| Component with attention selection | Encoder only | Decoder only | Encoder+Decoder |
|:---:|:---:|:---:|:---:|
| Group strategy | 42.2 | 46.2 | 40.0 |
| Subset strategy | 45.4 | 47.5 | 44.7 |

As is shown in Table 9, attention selection in only encoder (c.f. column "Encoder only") or decoder (c.f. column "Decoder only") would increase WER in comparison with the model with attention selection in both encoder and decoder (c.f. column "Encoder+Decoder"). We also note that attention head selection in encoder achieves lower WER than selection in decoder.