# OpenReview forum: "Pay Better Attention to Attention: Head Selection in Multilingual and Multi-Domain Sequence Modeling"
_NeurIPS.cc/2021/Conference — NeurIPS 2021 Poster_

### Official Review · Reviewer_zLUP · 2021-07-07

**Rating:** 6
**Confidence:** 4

**Summary:**

This paper aims to mitigate the problem of negative interference when learning models across different languages and domains. In particular, it proposes an approach that learns to share or specialize attention heads based on two different strategies, namely subset, and group selection.  The evaluation of multi-lingual/domain settings shows that this approach outperforms standard transformers and adapters without increasing the number of parameters.

**Limitations And Societal Impact:**

The authors refer to some of the limitations of the proposed approach by exploring how the hyperparameter affects performance and mentioning aspects that were not explored in this study. I would suggest that the authors discuss the lack of conditioning on the context representations for their selection variables and previous work that has focused on learning to share components for multilingual settings to better position the work with respect to prior work. The discussion regarding potential negative societal impact is generic and could have been a bit more elaborate.

**Main Review:**

The paper proposes a simple approach that is effective in multilingual and multidomain settings.  These are challenging settings due to the problem of negative interference which has been observed in past studies; mitigating this issue has a big potential impact and should be of interest to the community.

Learning to share or specialize components of neural networks for multilingual tasks has been proposed in the past [1]. The proposed idea is somewhat less general since its novelty lies specifically in the latent variable model with the two selection strategies for attention heads in transformers. Another limitation of the idea is that the selection model is independent of the context since the estimator used for the selection process learns directly the logit parameters. This makes it potentially less flexible and more data-dependent/specific.

The technical exposition was mostly clear and easy to follow but the positioning with respect to prior was missing emphasis on studies that learn to share components for multilingual and multidomain settings.  This is also reflected in the evaluation which lacks comparison with such methods. Other than that the evaluation provides evidence about the superiority of the approach against standard transformers and adapter baselines.  One potential issue is that the reporting is not thorough and some details for judging the results are missing e.g. showing the number of parameters for each model, detailed scores, and standard deviation when averaging.  One interesting result is that the proposed approach outperforms adapters without requiring their additional parameters and the finetuning step. This makes the adaptation process easier.

Questions/comments:
- In line 97, it is unclear what "our model provides more attention heads" means. It looks like that the model selects a subset of the heads for each language or domain. Could the authors provide some clarification?

- In the subset strategy, if the model assigns non-zero values to the logits of less than H heads Eq. 5 could lead to a smaller dimension than expecting after concatenation. How do the authors deal with this potential issue?

- For the multilingual experiment in Table 1, could the authors report the scores for the individual languages? The differences look small and it is unclear if the improvement comes from a single language or if it is present in all examined pairs.

- I would suggest a more clear reporting of the number of parameters that are trainable for each model and task (including the baseline transformer).  Currently, only the parameters of adapters and the proposed model are reported for the MT task (lines 214-215).

- This notation used for the logit parameters \phi^{(h)}_t is a bit counterintuitive. The function takes as argument 0 or 1 which is strange for indicating that the logits are specific to the outcome 0 or 1 of the target variable. Also, there is a subscript missing from the denominator in Eq. 4: \phi^{(h)} should be \phi^{(h)}_t.

[1] Share or Not? Learning to Schedule Language-Specific Capacity for Multilingual Translation, ICLR 2021

**Time Spent Reviewing:**

3.5

---

> ### Author Response · Authors · 2021-08-10
> **Author Response**
>
> We thank the reviewer for the detailed comments and constructive suggestions.
>
> 1. Clarification of “more attention heads”. In our formulation, a total of H' attention heads are provided in each layer, and each task (a language or a domain) would select a subset of H (H < H') heads so that only these H heads are activated in computation. It is possible that different attention heads are chosen by different tasks, and our approach is to learn how to select attention heads in order to mitigate interference between languages or domains and improve sequence model performance.
>
> 2. Dimension consistency for subset strategy. The reviewer is correct that a fixed number of H heads need to be selected in order to keep the dimension of attention outputs unchanged. In Eq. (5), we rank heads based on their likelihood of being selected by a given task t, and top H heads are chosen by the subset strategy for the task.
>
> 3. BLEU of individual languages. We report the language-level BLEU for O2M machine translation below. In comparison with Transformer, we observe consistent gains of group strategy (>=1 BLEU gain in 5 language directions, and >=0.5 in 9 language directions). In comparison with the adapter model, group strategy shows gains in 7 directions, and falls behind in the other 7 directions.
>
> Language | cs | de | es | et | fi | fr | gu | kk | lt | lv | ro | ru | tr | zh
>
> Transformer | 11.9 | 22.5 | 29.2 | 15.9 | 17 | 40.4 | 0.6 | 30.9 | 1.1 | 11.5 | 33.2 | 23.5 | 17.6 | 26.6
>
> Adapter        | 13.3 | 23.4 | 29.8 | 17 | 18 | 41.2 | 0.6 | 31.8 | 1.2 | 12.5 | 34.2 | 24.7 | 17.8 | 27.7
>
> Group strategy | 13.8 | 23.8 | 30.2 | 16.7 | 17.4 | 41.1 | 0.5 | 31.3 | 1.7 | 12.4 | 33.2 | 25.1 | 18 | 28.5
>
> 4.  Trainable parameter sizes. We thank the reviewer for suggesting the comparison, and report the total number (Million) of parameters in different models across tasks below. We will also include them in the revision.
>
> Task | Transformer | S2T Transformer | Adapter | Group strategy | Subset strategy
>
> MT (O2M/M2O) | 416 | - | 460 | 420 | 420
>
> Multilingual ASR | - | 311 | 501 | 346 | 346
>
> Multi-domain ASR | - | 321 | 392 | 356 | 356
>
> Multilingual ST | - | 311 | 54.8 | 346 | 346
>
> Multi-domain ST | - | 321 | 392 | 356 | 356
>
>
> 5. Logit parameter \phi^{(h)}. We apologize for the typo in Eq. (4), and the reviewer is correct that $\phi^{(h)}$ in the denominator should be $\phi^{(h)}_t$. The arguments 1 and 0 correspond to the binary decision on the selection of an attention head, and logits are used to estimate the likelihood of a head being selected in Eq. (4).

---

> > ### Comment · Reviewer_zLUP · 2021-08-12
> > **Response to rebuttal**
> >
> > Thank you for answering my questions and providing clarifications.
> >
> > I still think that better discussion and positioning with respect to studies that learn to share components for multilingual and multidomain settings are needed. The additional results provided and the transparency about the number of parameters will help the reader for the interpretation of the results if included in the final version. Hence, I decided to increase my score.

---

### Official Review · Reviewer_NgLe · 2021-07-08

**Rating:** 7
**Confidence:** 3

**Summary:**

A neat paper with an interesting idea for multi-task/multi-domain sequence learning.

**Limitations And Societal Impact:**

I expect to see more discussions on limitations.

**Main Review:**

The paper proposes a new approach for multi-task/multi-domain sequence learning where domain/task specific knowledge is represented via different head sets. Empirical results on multiple benchmarks of machine translation, speech recognition, and speech translation demonstrate the effectiveness of the proposed method.

The paper is very well written and the proposed method, at least to me, contains something new. Empirical studies are fairly rich with a few interesting insights disclosed.

Some of my concerns:
(1) If the method also works on tasks other than sequence learning, for example, classification and matching.
(2) On machine translation, the method is more useful on high resource languages, any reasons? If we still expect good performance on low resource languages, what can we do?
(3) Improvement over Adapter on MT seems not that significant, though the authors explain that their models are advantageous on efficiency. It would be great if the authors could show some quantitative results on efficiency comparison. That would be more convincing.




**Time Spent Reviewing:**

2 hours

---

> ### Author Response · Authors · 2021-08-10
> **Author Response**
>
> We thank the reviewer for the detailed comments and constructive suggestions.
>
> 1. Other tasks other than sequence learning. Thanks for suggesting classification and matching tasks. It would be our future work to extend the idea of attention selection beyond the text generation tasks.
>
> 2. Gains of machine translation in high- and low-resource languages. Existing studies [1] point out the interference between high- and low-resource languages, as the gains of low-resource performance is at the cost of high-resource performance. Attention head selection mitigates the interference, which would reduce the knowledge transfer to low-resource languages at the same time. Therefore, we observe that the gains in high-resource languages are more obvious.
>
> It would be of independent research interest to explore how to further enhance the performance of low-resource languages. For example, model could learn better head selection strategies by leveraging external knowledge such as language proximity and language resource size.
>
> [1] On Negative Interference in Multilingual Models: Findings and A Meta-Learning Treatment. Zirui Wang, Zachary C. Lipton, Yulia Tsvetkov. EMNLP 2020.
>
> 3. Efficiency metric. We report the decoding speed below, which is the number of decoded tokens per second (tokens/s) on a single GPU, as a quantitative metric of model inference efficiency. We would also include this metric in our revision.
>
> Task | Transformer | S2T Transformer | Adapter | Group strategy | Subset strategy
>
> MT (O2M) | 1140.3 | - | 1020.6 | 1136.8 | 1132.5
>
> MT (M2O) | 1251.8 | - | 1116.9 | 1244.7 | 1250.2
>
> Multilingual ASR | - | 1118.2 | 1016.4 | 1107.1 | 1113.6
>
> Multi-domain ASR | - | 1408.9 | 1274.3 | 1399.8 | 1402.9
>
> Multilingual ST | - | 1038.4 | 825.8 | 1024.4 | 1033.3
>
> Multi-domain ST | - | 1148.0 | 939.5 | 1140.6 | 1144.7

---

> > ### Comment · Reviewer_NgLe · 2021-08-15
> > **Response to Rebuttal**
> >
> > Thanks for the answers. It is a little disappointing that the proposed model does not show a significant advantage over Adapter on efficiency. I also expect the authors to further explore how to enhance the performance of the model on low-resource settings. After all, this is, to the best of my knowledge, a more important topic in MT research now.

---

### Official Review · Reviewer_HGp8 · 2021-07-14

**Rating:** 5
**Confidence:** 5

**Summary:**

Multi-head attention is an essential component for popular Transformer models. This paper proposes to learn shared and specialized attention heads for different languages and domains. The authors formulate attention selection as latent variables and adopt Gumbel softmax to select attention heads.  Experiments on text-to-text and speech-to-text translation verify the effectiveness of the proposed approach.

**Limitations And Societal Impact:**

It would be better to carefully polish the formulation and add more comparisons with dynamic networks, like Sparse-MoE.

**Main Review:**

The key idea is simple and intuitive. The writing is clear, and experiment settings are described in detail.

The main formulation in Section 3.2 is not very rigorous, even wrong. In Equation 2, the authors say that "can be computed by marginalizing over the posterior of latent variable z". According to line 122, $\Theta$ is parameters is Transformer.  It is important to note that p(z|\Theta)  is not posterior distribution. p(z|x,y) is the posterior distribution.  Also, p(z|\Theta) is usually written in p_{\theta}(z).   In lines 128-130, the authors say that "Specifically, we learn an inference network q(z) to approximate the true distribution p(z) and optimize the evidence lower bound (ELBO)". In the standard variational bound solution, q(z) here should be the posterior distribution q(z|x,y). However, the authors do not include $y$ in the implementation to compute $q(z|x,y)$.

The proposed approach achieves promising results on speech recognition.  However, on multilingual Machine Translation, the improvements achieved by the proposed approach are marginal.  Table 1 shows that the proposed approach only outperforms Adapter with 0.1 and 0.2 average  BLEU improvements.

Many related studies are missing. The proposed approach is highly related to dynamic networks and conditional computation.  Dynamic networks are networks that only use a part of architecture for inference. One representative model is Switch-Transformer. The authors should include these studies in the new version and add more comparisons on experiments.

**Time Spent Reviewing:**

4

---

> ### Author Response · Authors · 2021-08-10
> **Author Response**
>
> We thank the reviewer for the detailed comments and constructive suggestions.
>
> 1. Latent variable z. We want to clarify the misunderstanding on how the latent variable z is defined in our work, and distinguish it from the latent variable  in Variational Neural Machine Translation (VNMT) [1]. Our work introduces latent variable z to modulate the selection of attention heads based on a given task t (the language or domain). Let’s explain Eq. (2) in the simplest case that there is only one task, i.e., all pairs of (x, y) correspond to the same task t and have the same subset of attention heads. As described in lines 124-126, a variable $z_{t}$ selects attention heads within a Transformer. Hence the selection is related with parameters $\Theta$ instead of x or y. This is why the posterior probability of z is conditioned on $\Theta$ in our formulation. Moreover, z is a discrete variable with Bernoulli distribution in our work.
>
> Latent variables in VNMT model the underlying semantic space of inputs for translation [1]. Therefore in VNMT, the prior probability of latent variables is conditioned on x, and the posterior is conditioned on x and y. It is often assumed that latent variables follow a continuous distribution such as Gaussian distribution, which is different from the discrete variable in our work.
>
> [1] Variational Neural Machine Translation. Biao Zhang, Deyi Xiong, Jinsong Su, Hong Duan, Min Zhang. EMNLP 2016.
>
> 2. Marginal gains over adapter in machine translation. Despite similar performance to the adapter model in MT, the attention head selection approach is advantageous in terms of smaller model size and better inference efficiency. We compare the total number (Million) of model parameters below.
>
> Task | Transformer | Adapter | Group strategy | Subset strategy
>
> MT (O2M/M2O) | 416 | 460 | 420 | 420
>
> To compare the computation efficiency, we report the decoding speed below, which is the number of decoded tokens per second (tokens/s) on a single GPU.
>
> Task | Transformer | Adapter | Group strategy | Subset strategy
>
> MT (O2M) | 1140.3 | 1020.6 | 1136.8 | 1132.5
>
> MT (M2O) | 1251.8 | 1116.9 | 1244.7 | 1250.2
>
> 3. We thank the reviewer for suggesting related studies on dynamic networks and conditional computation. We included sparse conditional models such as GShard in related works, and would add Switch Transformer and other recent approaches for the completeness of discussion.
> We note some key differences of our work from GShard and Switch Transformer: (1) we focus on attention head selection while GShard and Switch Transformer choose an FFN module from a mixture of FFN experts; (2) our head selection is conditioned on the language or the data domain, while GShard and Switch Transformer choose FFN for each input token. Our study is orthogonal to these two Mixture-of-Experts models in that the different modules (attention heads vs. FFN) are selected using different strategies (task-based vs. token-based strategy). It is of independent interest to adapt the FFN selection to attention head selection, and thus beyond the scope of this paper.

---

> > ### Comment · Reviewer_HGp8 · 2021-08-16
> > **Conditional VAE**
> >
> > Thanks for the responses!
> >
> > There may be some misunderstandings on my questions. I want to say that the authors use the unconditional latent variable framework to formulate conditional latent variable tasks, which does not make sense. VNMT is a kind of conditional latent variable formulation, but not the only one. The authors should use a conditional latent variable framework, rather than VNMT, to formulate the mentioned task.
> >
> > As the authors mention,  the selection of attention heads is based on the given task $t$.  $t$ is the conditional label, and the output sequence is $y$. If we do not consider tasks, instead only suppose that the sequence $y$ is generated via latent variable $z$. That's the settings of the unconditional latent variable framework. Eq.2 and Eq.3 follow the original formulations of VAE (unconditional latent variable framework) from [1].  In the conditional formulation, at least the conditional label should be described in notations to distinguish posterior distribution and prior distribution.
> >
> > In all, the authors should follow CVAE formulations rather than VAE formulations. I decide to keep my score unchanged.
> >
> >
> >
> >
> > [1] Auto-Encoding Variational Bayes.

---

> > > ### Author Response · Authors · 2021-08-19
> > > **Response to Conditional VAE**
> > >
> > > We thank the review for the clarification. As reviewer mentioned, Eq. (2) and (3) formulate the head selection in sequence modeling given a single task, and we use a universal latent variable $z$ in these formulations. We later introduce task-dependent variables $\\{z_{t}\\}$ to discuss the task-dependent head selection.
> > > It is straightforward to extend Eq. (2) and (3) from the single-task to multi-task setting. Suppose that the data is $D=\\{(x, y)\\}$, and the set of data corresponding to task $t$ is $D_{t}=\\{(x_{t}, y_{t})\\}$. In multi-task setting, Eq. (2) is re-written as:
> > >
> > > $p(y|x,\Theta)=\prod\limits_{t}E_{p(z_t|\Theta)}[p(y_{t}|x_{t},z_{t})]$.
> > >
> > > Similar to Eq.(3), we can again apply the evidence lower bound to multi-task setting:
> > >
> > > $\sum\limits_{t}\log p(y_{t}|x_{t}) \geq \sum\limits_{t} \left\(E_{q_{\phi} (z_{t})}[\log p_{\theta}(y_{t}|x_{t}, z_{t})] - \text{KL}(q_{\phi}(z_{t}) || p(z_{t}))\right)$.
> > >
> > > Therefore the training and inference would not change in multi-task setting. We apologize for the confusion here, and would provide the multi-task formulations corresponding to Eq. (2) and (3) with variables $\{z_{t}\}$.

---

> > > > ### Comment · Reviewer_HGp8 · 2021-08-30
> > > > **Response**
> > > >
> > > > The authors do not give a convincing explanation. The authors claim that variables are task-dependent. I believe that the authors also treat it as a conditional variable in implementation. What I am concerned about is the formulation. The authors seem to muddle the difference between conditional VAE and unconditional VAE.

---

> > > > > ### Author Response · Authors · 2021-08-31
> > > > > **Response to the conditional formulation**
> > > > >
> > > > > Following the reviewer’s suggestion, we provide the conditional formulation to reduce the confusion of defining variable z. Given the task t (a language or a domain in multilingual and multi-domain settings respectively), we write the posterior of $p(y|x)$:
> > > > >
> > > > > $p(y|x, \Theta, t) = \int p(y|x, z, t)p(z|t, \Theta) dz$.
> > > > >
> > > > > The corresponding ELBO is
> > > > >
> > > > > $\log p(y|x, t) \geq E_{q_{\phi}(z|t)}[\log p_{\theta}(y|x,z,t)] - KL(q_{\phi}(z|t) || p(z|t))$.
> > > > >
> > > > > In comparison with existing conditional variational models [1] and [2], the main difference is that the task t is given as prior knowledge and the variable z, which selects attention heads based on the task, has a prior directly conditioned on t instead of x.
> > > > >
> > > > > Suppose that the sample $(x_t, y_t)$ belongs to the task $t$, and variable $z_{t}$ has the distribution conditioned on $t$. The bound can be rewritten as
> > > > >
> > > > > $\log p(y_t|x_t) \geq E_{q_{\phi}(z_t)}[\log p_{\theta}(y_t|x_t, z_t)] - KL(q_{\phi}(z_t) || p(z_t))$
> > > > >
> > > > > This is the same as the formulation given in our last response.
> > > > >
> > > > > [1] Wei Y, Zhang Z, Wang Y, Xu M, Yang Y, Yan S, Wang M. DerainCycleGAN: Rain attentive CycleGAN for single image deraining and rainmaking. IEEE Transactions on Image Processing. 2021 Apr 30;30:4788-801.
> > > > >
> > > > > [2] Calixto I, Rios M, Aziz W. Latent Variable Model for Multi-modal Translation. In Proceedings of the 57th Annual Meeting of the Association for Computational Linguistics 2019 Jul (pp. 6392-6405).

---

> > > > > > ### Comment · Reviewer_HGp8 · 2021-09-03
> > > > > > **Response**
> > > > > >
> > > > > > Thanks for the responses.
> > > > > >
> > > > > > Here is the format of unconditional VAE:
> > > > > >
> > > > > > l_{i,\theta} = E_{z \sim q_{\theta}(z|x)}logp(x|z) + KL(q_{\theta}(z|x_i) || p(z))
> > > > > >
> > > > > > If the authors know how ELBO works, they must know that q is the posterior distribution concerning the target variable x.  Only this, we can get the ELBO format via a sequence of deductions.
> > > > > >
> > > > > >
> > > > > > The target of this paper is to model y given x and t.  To get ELBO, q should be the posterior distribution concerning the target variable y, and input x and t.  Only this we can get the right ELBO. That's how previous conditional VAE models work, including papers mentioned by the authors.
> > > > > >
> > > > > > I am shocked by the formulation in this paper.  The authors define q as the posterior distribution with respect to $t$.  It does not make sense to get the ELBO given posterior distribution z without y and posterior distribution y with z.
> > > > > >
> > > > > >
> > > > > > I will keep my scores until the authors give a strict derivation process.

---

> > > > > > > ### Author Response · Authors · 2021-09-03
> > > > > > > **Derivation of the bound**
> > > > > > >
> > > > > > > Firstly, we want to clarify an important assumption that the prior of latent variable z is conditioned on the task t regardless of x and y. Taking multilingual setting as an example. The interference is between different languages, and what is proposed in this work is to select attention heads based on the language instead of each data sample.
> > > > > > >
> > > > > > > Given a task t, suppose that the corresponding samples are {$(x^{1}, y^{1}), \ldots, (x^{i}, y^{i}), \ldots, (x^{n}, y^{n})$}. All of these samples in the same task would select the same set of attention heads. That being said, we have $p(z|x^{1},t)=\cdots=p(z|x^{i},t)=\cdots=p(z|x^{n}, t)$. Therefore, the prior $p(z|x,t)$ can be simplified as $p(z|x,t)=p(z|t)$.  Similarly, we can simplify the posterior: $q_{\phi}(z|x,y,t)=q(z|t)$.
> > > > > > >
> > > > > > > We derive the bound below:
> > > > > > >
> > > > > > > $E_{q_{\phi}(z|t)}[\log p_{\theta}(y|x,z,t)] - KL(q_{\phi}(z|t) || p(z|t))$
> > > > > > >
> > > > > > > $= E_{q_{\phi}(z|t)}[\log p_{\theta}(y|x,z,t)] - E_{q_{\phi}(z|t)}[\log q_{\phi}(z|t) - \log p(z|t)]$
> > > > > > >
> > > > > > > $= E_{q_{\phi}(z|t)}[\log p_{\theta}(y|x,z,t) + \log p(z|t) - \log q_{\phi}(z|t)] $
> > > > > > >
> > > > > > > $\leq \log E_{q_{\phi}(z|t)}[p_{\theta}(y|x,z,t) * p(z|t) / q_{\phi}(z|t)]$                (based on Jensen's inequality)
> > > > > > >
> > > > > > > $\leq \log \int p_{\theta}(y|x,z,t) * p(z|t) dz$
> > > > > > >
> > > > > > > $\leq \log p(y|x, t)$.
> > > > > > >
> > > > > > > Therefore we have
> > > > > > >
> > > > > > > $\log p(y|x, t) \geq E_{q_{\phi}(z|t)}[\log p_{\theta}(y|x,z,t)] - KL(q_{\phi}(z|t) || p(z|t)).$

---

> > > > > > > > ### Comment · Reviewer_HGp8 · 2021-09-07
> > > > > > > > **Response**
> > > > > > > >
> > > > > > > > Thanks for the clarification! The authors use p(z|t) to replace p(z|x,y,t) or p(z|x,t) to get ELBO, which makes the derivation clearer.
> > > > > > > >
> > > > > > > > The authors addressed my concerns after giving a detailed derivation process. However, this paper still have other unclear points, and it would be better to add more explanations.  Compared to other dynamic networks which have applied dynamic operations on machine translation, e.g, [1],  the contribution of this paper is a little bit limited from my perspective. Therefore, I decide to raise my score to 5.
> > > > > > > >
> > > > > > > >
> > > > > > > > [1] Dynamic deep neural networks: Optimizing accuracy-efficiency trade-offs by selective execution.

---

### Official Review · Reviewer_WC28 · 2021-07-19

**Rating:** 6
**Confidence:** 3

**Summary:**

The paper proposes a model of latent selection of attention heads to improve the performance of sequence modelling in multilingual and multi-domain NMT, speech recognition, and speech-to-text translation.

**Limitations And Societal Impact:**

The authors have adequately addressed the limitations and potential negative societal impact of their work.

**Main Review:**

_Originality_: The methods proposed in the paper are novel and well distinguished from related work.

_Quality_: The method proposed is interesting. Experiments were nicely conducted on multiple tasks to show the better performance of the model on different settings when compared to the baselines. However, all the experiments lack error bars over different random seeds (which is different from averaging checkpoints of the same run, or showing the average of the results over all languages). Since oftentimes the difference between the model proposed and the baseline is small, it would be important to show these error bars to better understand how much the proposed method improves upon the baselines. Furthermore, the method itself could be better explained and I wish more space was given to §3. Particularly, what is the Variational NMT model used? Is there a learnable prior?

_Clarity_: The paper is well-written and well-structured, albeit explanations of the methods could be more detailed as mentioned above.

_Significance_: The ideas and results shown in this work are relevant to researchers and practioners that study multilingual and multi-domain models. The paper is extensive in the number of tasks in which they test their model and baselines in and that can provide insight into future ideas.

**Time Spent Reviewing:**

2h

---

> ### Author Response · Authors · 2021-08-10
> **Author Response**
>
> We thank the reviewer for the detailed comments and constructive suggestions.
>
> 1. Error bars over different random seeds. Thanks for suggesting experiments with different seeds. Head selection with group strategy achieves >1.0 BLEU gain over both S2T Transformer and adapter in speech translation task. For machine translation and ASR tasks, the gains of head selection are relatively small over adapter models. We would launch experiments with different seeds to better compare the model performance in these two tasks.
>
> 2. Explanation of the model.
> Our work introduces latent variable z to modulate the selection of attention heads based on a given task (a language or a domain). Latent variable z is discrete and is assumed to follow Bernoulli distribution. By assuming that each head is selected with identical probability, the prior is set as $p(z=1)=\frac{H}{H’}$, where H is the number of selected heads and H' is the total number of heads.
>
> We want to clarify the difference from previous studies on Variational NMT (VNMT) despite the fact that both our work and VNMT introduce latent variables to translation models. Latent variables in VNMT model the underlying semantic space of the inputs for translation [1]. Therefore in VNMT, the prior probability of latent variables is conditioned on the input x. It is assumed that the latent variable has continuous distribution, which is different from the discrete latent variable in our work.
>
> [1] Variational Neural Machine Translation. Biao Zhang, Deyi Xiong, Jinsong Su, Hong Duan, Min Zhang. EMNLP 2016.

---

> > ### Comment · Reviewer_WC28 · 2021-08-13
> > **Response to rebuttal**
> >
> > Thank you for your clarifications. I decided to keep my score.

---

### Decision · Program_Chairs · 2021-09-28

**Decision:**

Accept (Poster)

**Comment:**

This paper analyzes multi-head attention in multilingual and multi-domain sequence modeling tasks. The paper claims that non-selective attention sharing is sub-optimal for achieving good generalization across all languages and domains, and proposes new attention sharing strategies for different languages and domains to mitigate interference. Experiments are reported in speech recognition, text-to-text and speech-to-text translation, with consistent gains.

Most reviewers agree that the proposed method is novel and interesting and the paper is clear and well written, though some experiments need to be clarified (addressed in the rebuttal). The main weaknesses pointed out by the reviewers is a non-standard conditional VAE formulation, which the authors clarified I their rebuttal, and the lack of discussion with and positioning with respect to studies that learn to share components for multilingual and multidomain settings (including adaptors). In a future iteration, I urge the authors to take into account the detailed comments made by the reviewers when preparing a new version of their paper.

**Consistency Experiment:**

NeurIPS has a long history of experimentation. In 2014, NeurIPS ran an experiment in which 10% of submissions were reviewed by two independent committees to quantify the randomness in the review process. This year, we repeated a variant of this experiment to see how the quality of the review process has changed over time.  This paper was part of the experiment and was therefore assigned to two committees (consisting of reviewers, an Area Chair, and a Senior Area Chair) that reached independent decisions.  If both committees made the same recommendation, this recommendation was followed. If a single committee recommended acceptance, the paper was accepted (with the exception of a few cases in which the other committee identified what we considered a fatal flaw, e.g., an error in a key result).

This copy’s committee reached the following decision: **Reject**

The other committee assigned to the paper recommended **Accept (Poster)**.  You can find the other set of reviews, along with any follow up discussion with the authors here:
https://openreview.net/forum?id=c7VY6-YKek2